# MoreauPruner: Robust Structured Pruning of Large Language Models against Weight Perturbations

## Abstract

In the existing model pruning literature, the weight gradient has been extensively utilized to measure the importance of weight, where the gradient is well-known to be sensitive to perturbations. On the other hand, the widely used large language models (LLMs) have several billion model parameters, which could increase the fragility of few-shot gradient pruning. In this work, we experimentally show that one-shot gradient pruning algorithms could lead to unstable results under perturbations to model weights. Even the minor error of switching between data formats bfloat16 and float16 could result in obviously different outcomes. To address such instabilities, we leverage optimization analysis and propose an LLM structural pruning method, called MoreauPruner, with provable robustness against weight perturbations. In MoreauPruner, the model weight importance is estimated based on the neural network's Moreau envelope, which can be flexibly combined with $\ell_1$-norm regularization techniques to induce the sparsity required in the pruning task. We extensively evaluate the MoreauPruner algorithm on several well-known LLMs, including LLaMA-7B, LLaMA-13B, LLaMA3-8B, and Vicuna-7B. Our numerical results suggest the robustness of MoreauPruner against weight perturbation and how robust importance estimation in MoreauPruner contributes to successful accuracy-based scores compared to several existing pruning methods.

## 1 Introduction

In the rapidly evolving field of Natural Language Processing (NLP), transformer-based Large Language Models such as GPTs (Dettmers et al., 2022) and LLaMAs (Touvron et al., 2023; AI@Meta, 2024) have become foundational technologies, driving significant advances across various tasks. These models excel in understanding and generating human language due to their scalable architecture, which allows performance to improve with an increase in parameters. However, deploying these large models poses significant challenges due to their substantial computational and memory demands. To address these challenges, considerable research has been directed toward model pruning (Han et al., 2015; Wen et al., 2016; Ma et al., 2023; Zhang et al., 2023), a technique aimed at reducing model size while maintaining or enhancing model performance.

While effective in accelerating LLMs for efficient deployment, existing pruning methods generally focus on fixed pre-trained models, neglecting potential perturbations in the weights and their effect on pruning outcomes. These perturbations can originate from various sources, including quantization errors during transitions between precision levels, errors introduced by post-training operation merging(DeepSeek-AI, 2024), position embedding extension(Su et al., 2024) when the attention window is enlarged and so on. With minor changes mentioned above, the modified models usually produce similar outputs compared with unmodified models. Therefore, when there is a need to prune those slightly modified models, we may expect that the pruned modified models are similar to the pruned original models. For example, some popular LLMs (Touvron et al., 2023) are trained with the weight format bfloat16 (BF16) and deployed with the weight format float16 (FP16). As both BF16 and FP16 utilize 16-bit to represent a floating point, the negligible transition error will not affect inference results in most cases. Considering that the basic idea of pruning is removing unnecessary weights and keeping the essential weights, it is straightforward to believe that the models pruned from BF16 and FP16 will be close to each other. However, current gradient-dependent pruning methods (Ma et al., 2023; Zhang et al., 2023; LeCun et al., 1989; Hassibi & Stork, 1992) utilize gradient to

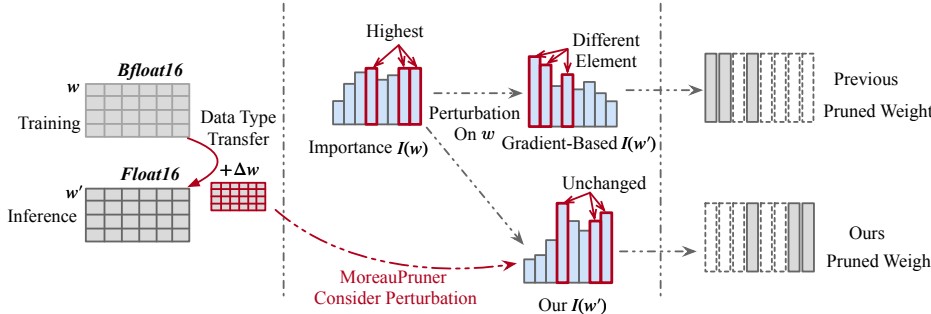

Figure 1: While gradient-based pruning methods are sensitive to weight perturbation, the proposed MoreauPruner gives a robust estimation of weight importance.

indicate the importance of weight elements while gradient is known to be sensitive to such weight perturbations, leading to significant variations in pruning outcomes, as depicted in Figure 1. Such inconsistency in pruned outcomes could be anti-intuitive, and a robust weight importance estimation against weight perturbation is intuitively beneficial to enhance the performance of pruning algorithms.

This paper introduces MoreauPruner, a novel robust structural pruning algorithm for LLMs, designed to mitigate the effects of weight perturbations while preserving model performance. MoreauPruner utilizes the gradient of the loss function's Moreau envelope (Moreau, 1965; Zhang & Farnia, 2023; T Dinh et al., 2020), a well-established optimization tool for function smoothing, to reduce weight sensitivity to perturbations during the pruning process. We show that the gradient of the Moreau envelope remains stable within the neighborhood of given weight in parameter space. This stability enables MoreauPruner to generate robustness pruning result against weight perturbations, with any norm-bounded perturbation resulting in only a bounded change of the Moreau gradient. Additionally, by incorporating an $\ell_1$-group-norm-based regularization penalty, MoreauPruner promotes group-level sparsity in the gradient, which is suitable for structural pruning to facilitate real-life acceleration on hardware platforms. Our empirical results suggest that MoreauPruner improves the robustness of pruning outcomes against weight perturbations and achieves state-of-the-art post-pruning model performance among baseline LLM pruning methods.

Our contributions through this work are threefold:

- We emphasize the importance of consistent pruning criteria against minor weight perturbations, an aspect previously neglected in the literature. This work is among the first to tackle the robustness of pruning algorithms to such perturbations.

- We introduce MoreauPruner, a structural pruning algorithm that offers provable robustness to weight perturbations, leveraging the Moreau envelope to ensure the smoothness and stability of the pruning process.

- Through extensive experimentation with widely-used large language models such as LLaMA-7B, LLaMA-13B, LLaMA3-8B, and Vicuna-7B, we demonstrate that MoreauPruner achieves a state-of-the-art performance in both robustness to weight perturbation and overall performance of compressed models.

## 2 RELATED WORK

### 2.1 EFFICIENT LARGE LANGUAGE MODELS

Large Language Models (LLMs) (Touvron et al., 2023; Achiam et al., 2023; Chiang et al., 2023; AI@Meta, 2024) have achieved remarkable performance by following the scaling laws (Kaplan et al., 2020). However, deploying LLMs can be challenging due to high inference costs in resource-limited scenarios. Various methods have been proposed to reduce model size, including knowledge distillation (Hinton et al., 2015; Sanh et al., 2019; Sun et al., 2019; 2020), which involves transferring the knowledge from the original model to a smaller one; model quantization (Dettmers et al., 2022; Xiao et al., 2023; Yao et al., 2022; Zafrir et al., 2019), which reduces the bit length of the model weights; and model pruning (Han et al., 2015; Frankle & Carbin, 2018; Fang et al., 2023; Park et al., 2023), which involves removing non-essential weights to speed up inference. This work primarily

focuses on pruning LLMs (Xia et al., 2023; Ma et al., 2023; Bair et al., 2024; Xu & Zhang, 2024; Ashkboos et al., 2024; Zhang et al., 2024; Yin et al., 2023; Ji et al., 2023; van der Ouderaa et al., 2023; Dong et al., 2024), where gradients are particularly sensitive to weight perturbations due to the large scale of the model.

## 2.2 PRUNING CRITERIA

To determine which weights to prune during the pruning phase, the importance of each weight is assessed using various criteria. Several studies (Sun et al., 2023; Li et al., 2018; Han et al., 2015; Elesedy et al., 2020) adopt a magnitude-based criterion, retaining weights with larger magnitudes post-pruning. Recent approaches (Sun et al., 2023) also consider activation values to evaluate weight importance. Some prevalent criteria are based on Taylor Expansion approximation (LeCun et al., 1989; Ma et al., 2023; Yu et al., 2022; Hassibi et al., 1993; Hassibi & Stork, 1992), utilizing differential information (zero-th, first, and second order) to estimate output changes if weights are removed. Notably, (Zhang et al., 2023) highlights that, in LLMs, gradients can be efficiently approximated using low-rank methods (Hu et al., 2021) when the direct computation of the gradient is too costly. Nevertheless, gradients can be highly sensitive to weight modifications, rendering gradient-based pruning criteria susceptible to variations in weight. In response, MoreauPruner offers proven robustness against any norm-bounded weight perturbation, maintaining model performance.

## 3 PRELIMINARIES

In this section, we provide a review of prior Taylor-expansion-based structural pruning methods, along with the notation and definitions used in the paper.

## 3.1 NOTATION AND DEFINITIONS

Let $\mathcal{D} = \{\boldsymbol{x}_i\}_{i=1}^N$ denotes a text dataset with N samples. $f(\boldsymbol{w}, \boldsymbol{x})$ is the next token prediction loss on sample $\boldsymbol{x}$ with a parameterized language model and its weight $\boldsymbol{w} \in \mathbb{R}^d$. Then the expectation of loss on dataset $\mathcal{D}$ is

$$L(\boldsymbol{w}, \mathcal{D}) = \frac{1}{N} \sum_{i=1}^N f(\boldsymbol{w}, \boldsymbol{x}_i). \tag{1}$$

The $\ell_p$-norm of an input vector $\mathbf{v}$ is represented as $\|\mathbf{v}\|_p$. Furthermore, we use notation $\|\mathbf{v}\|_{p,q}$ to denote the $\ell_{p,q}$-group-norm of $\mathbf{v}$ defined in the following equation for given variable subsets $S_1, ..., S_t \subset \{1, ..., d\}$:

$$\|\mathbf{v}\|_{p,q} = \big\| [\|\mathbf{v}_{S_1}\|_p, ..., \|\mathbf{v}_{S_t}\|_p] \big\|_q, \tag{2}$$

which means $\|\mathbf{v}\|_{p,q}$ is the $\ell_q$-norm of a vector containing the $\ell_p$-norms of the subvectors of $\mathbf{v}$ characterized by index subsets $S_1, ..., S_t \subset \{1, ..., d\}$.

## 3.2 ESTIMATING IMPORTANCE SCORE VIA TAYLOR EXPANSION

Recent pruning methods usually estimate the impact of removing the $k$-th element of parameter vector $\boldsymbol{w}$ via Taylor Expansion,

$$I(\boldsymbol{w}^{(k)}) = \|L(\boldsymbol{w}, \mathcal{D}) - L(\boldsymbol{w}^{(k)} = 0, \mathcal{D})\|_1$$
$$= \left\| \frac{\partial L(\boldsymbol{w}, \mathcal{D})}{\partial \boldsymbol{w}^{(k)}} \boldsymbol{w}^{(k)} - \frac{1}{2} \boldsymbol{w}^{(k)} \mathbf{H}_{kk}(\mathbf{x}) \boldsymbol{w}^{(k)} + \mathcal{O}(|\boldsymbol{w}^{(k)}|^3) \right\|_1 .$$

In the above equations, $L(\boldsymbol{w}^{(k)} = 0, \mathcal{D})$ denotes masking out a single weight element $\boldsymbol{w}^{(k)}$ in the neural network. Hessian matrix $\mathbf{H}$ is approximated by a diagonal one and $\mathbf{H}_{kk}$ is the $k$-th element on the diagonal. In some of previous pruning methods, the first-term is typically neglected since the the model is well-trained and converged on the training dataset, where $\frac{\partial L(\boldsymbol{w}, \mathcal{D})}{\partial \boldsymbol{w}^{(k)}} \approx 0$. However, a recent LLM pruning work(Ma et al., 2023) point out the calibration dataset used in pruning is out of the training data and $\frac{\partial L(\boldsymbol{w}, \mathcal{D})}{\partial \boldsymbol{w}^{(k)}} \neq 0$. Given that the heavy computational cost of the Hessian matrix is unacceptable for LLMs, unlike small models, the importance score of parameter $\boldsymbol{w}^{(k)}$ can be approximated using the first term in the Taylor Expansion,

$$I(\boldsymbol{w}^{(k)}) = \left\| \frac{\partial L(\boldsymbol{w}, \mathcal{D})}{\partial \boldsymbol{w}^{(k)}} \boldsymbol{w}^{(k)} \right\|_1 . \tag{3}$$

Unstructured pruning algorithms directly remove weight elements with lower importance score. Contrast to them, to achieve real-time acceleration on hardware, structural pruning algorithms remove weight elements in group, *i.e.*, all elements in a channel, blocks or heads. The importance score of the weight vector in a structure $I(\boldsymbol{w}_i)$ can be easy obtained by summarizing the importance of its elements $I(\boldsymbol{w}_i^{(k)})$,

$$I(\boldsymbol{w}_i) = \sum_k I(\boldsymbol{w}_i^{(k)}). \tag{4}$$

Once the importance score of each structure $I(\boldsymbol{w}_i)$ is obtained, existing pruning algorithms tend to pruning structures with smaller importance scores. However, as we mentioned in Section 1, the simple gradient in Equation (3) is sensitive to weight perturbations, which further leads to an unstable pruning result. Motivated by this fact, we have designed a robust pruning criterion in MoreauPruner and detailed it in the next section. By substituting the simple gradient-based importance score in Equation (3) with this new criterion, we demonstrate that the pruning algorithm becomes more robust to weight perturbations and yields improved post-pruning performance.

### 3.3 DEPENDENCY-AWARE STRUCTURAL PRUNING

Previous works(Fang et al., 2023; Ma et al., 2023) suggest that structural pruning should consider dependency among structures. Here, a weight group $\mathcal{G} = \{\boldsymbol{w}_i\}_{i=1}^M$ represents a collection of coupled structures, where $M$ is the number of structures in one group, and $\boldsymbol{w}_i$ denotes the weight for each structure. The group can be effiently detected by Fang et al. (2023). And the importance score of the group $\mathcal{G}$ is then estimated as follows:

$$I(\mathcal{G}) = \underset{i=1}{\overset{M}{Agg}} I(\boldsymbol{w}_i), \tag{5}$$

where $Agg$ is a customized aggregation function chosen from options like Summation, Production, Max, etc. After assessing the importance of each group, groups with lower importance are pruned to achieve a pre-determined pruning ratio. We adopt the pruning strategy in our MoreauPruner and choose Summation in Equation (5), following Ma et al. (2023).

## 4 MOREAUPRUNER

In this section, we introduce the proposed pruning method, MoreauPruner. We start by detailing the proposed perturbation-robust pruning criteria. In the second subsection, we introduce the two versions of MoreauPruner.

### 4.1 ROBUSTIFYING GRADIENT VIA MOREAU ENVELOPE

Here we leverage the notion of Moreau envelope from the convex optimization literature to propose an optimization-based approach to robust gradient-based pruning. The considered robust gradient follows Moreau-Yosida regularization, based on which the Moreau envelope of a neural network's parameters is defined as follows.

**Definition 1.** *Consider function $g : \mathbb{R}^d \to \mathbb{R}$ and regularization parameter $\rho > 0$. The Moreau envelope of $g$ at input weight $\boldsymbol{w}$, $g^\rho : \mathbb{R}^d \to \mathbb{R}$ is defined to be*

$$g^\rho(\boldsymbol{w}) := \inf_{\tilde{\boldsymbol{w}}} g(\tilde{\boldsymbol{w}}) + \frac{1}{2\rho}\big\|\tilde{\boldsymbol{w}} - \boldsymbol{w}\big\|_2^2. \tag{6}$$

Following Zhang & Farnia (2023), instead of utilizing the simple gradient as previous pruning methods do, we employ the gradient of the Moreau envelope as a robust evaluation of the local sensitivity of the loss function sensitivity to altering the model weights,

**Definition 2.** *Given input weight $\boldsymbol{w}$ and regularization parameter $\rho > 0$, we define MoreauGrad as the Moreau envelope $g^\rho$'s gradient $\mathrm{MG}^\rho[g] : \mathbb{R}^d \to \mathbb{R}^d$:*

$$\mathrm{MG}^\rho[g](\boldsymbol{w}) := \nabla g^\rho(\boldsymbol{w}). \tag{7}$$

To analyze the gradient of Moreau envelope, we first discuss the optimization-based smoothing enforced by the Moreau envelope. Note that the Moreau envelope is known as an optimization tool to turn non-smooth convex functions (e.g. $\ell_1$-norm) into smooth functions, where the smoothness is usually regarding the input variable $\boldsymbol{x}$. Here in the pruning case, we discuss the smoothness regarding

the function parameters $\boldsymbol{w}$ and extend the result to the weakly-convex functions which also apply to non-convex functions.

**Theorem 1.** *Suppose that the parameterized function $g(\boldsymbol{w}) : \mathbb{R}^d \to \mathbb{R}$ is $\beta$-Lipschitz, i.e. it satisfies $\big|g(\boldsymbol{w}) - g(\mathbf{v})\big| \leq \beta\|\boldsymbol{w} - \mathbf{v}\|_2$ for every $\mathbf{v}, \boldsymbol{w} \in \mathbb{R}^d$. Consider the Gaussian-smoothed $g_\sigma(\boldsymbol{w}) = \mathbb{E}_{\mathbf{u}\sim\mathcal{N}(\mathbf{0},\sigma^2 I)}\big[g(\boldsymbol{w} + \mathbf{u})\big]$. Then, for every $0 < \rho < \frac{\sigma}{\beta}$, the following robustness guarantee will hold of the Moreau envelope of the Gaussian-smoothed $g_\sigma^\rho(\boldsymbol{w})$:*

$$\|\mathrm{MG}^\rho[g_\sigma](\boldsymbol{w}_1) - \mathrm{MG}^\rho[g_\sigma](\boldsymbol{w}_2)\|_2 \leq \frac{\sigma\|\boldsymbol{w}_1 - \boldsymbol{w}_2\|_2}{\min\{\sigma\rho, \sigma - \rho\beta\}}.$$

We defer the proof of the above theorem into appendices due to the space limitation.

Theorem 1 indicates that the change of gradient of Moreau Envelope will be bounded by the change of weights, denoting the robustness property of MoreauGrad $\mathrm{MG}^\rho[g_\sigma](\boldsymbol{w})$. We note that the above definition can be combined with sparsity-based norm penalties, such as $\ell_1$-norm $\|\cdot\|_1$ or $\ell_{2,1}$-group norm $\|\cdot\|_{2,1}$. Here, we generalize Zhang & Farnia (2023)'s definition of (group)sparse-Moreau envelop. Given a convex function $h : \mathbb{R}^d \to \mathbb{R}$, we propose the following definition of $h$-Moreau envelope:

**Definition 3.** *Given convex function $h$, input weight $\boldsymbol{w}$, and regularization parameter $\rho > 0$, we define $h$-MoreauGrad of function $g$, denoted by $h$-$\mathrm{MG}^\rho[g](\mathbf{w})$, as $\frac{1}{\rho}\big(\mathbf{v}^*(\mathbf{w}) - \mathbf{w}\big)$ where $\mathbf{v}^*(\mathbf{w})$ denotes the optimal solution to the following optimization problem:*

$$\min_{\mathbf{v}\in\mathbb{R}^d} \; g(\mathbf{v}) + \frac{1}{2\rho}\big\|\mathbf{v} - \mathbf{w}\big\|_2^2 + h\big(\mathbf{v} - \mathbf{w}\big). \tag{8}$$

Here, we extend the robustness guarantee of Theorem 1 to a general $h$-MoreauGrad.

**Theorem 2.** *Consider the setting of Theorem 1 and suppose $h$ is a convex function. Then, for every $0 < \rho < \frac{\sigma}{\beta}$, the following robustness guarantee will hold of the $h$-Moreau envelope of $g_\sigma^\rho(\boldsymbol{w})$:*

$$\|h\text{-}\mathrm{MG}^\rho[g_\sigma](\boldsymbol{w}_1) - h\text{-}\mathrm{MG}^\rho[g_\sigma](\boldsymbol{w}_2)\|_2 \leq \frac{\sigma\|\boldsymbol{w}_1 - \boldsymbol{w}_2\|_2}{\min\{\sigma\rho, \sigma - \rho\beta\}}.$$

We defer the proof of the above theorem into appendices due to the space limitation.

Similar to Theorem 1, Theorem 2 shows the robustness of $h$-MoreauGrad $h$-$\mathrm{MG}^\rho[g_\sigma](\boldsymbol{w})$ for a given convex function $h$. In our numerical analysis, we specifically focus on GroupSparse-MoreauGrad which is the $h$-MoreauGrad with the group-norm $h(\mathbf{v}) = \eta\|\mathbf{v}\|_{2,1}$. $\eta$ is the sparsity parameter.

---

**Algorithm 1** MoreauPruner Algorithm

---

**Require:** samples $\boldsymbol{x}$, network with parameter $f(\boldsymbol{w})$, regularization parameter $\rho$, group-sparsity $\eta$, noise std $\sigma$, stepsize $\gamma$, and optimization length $T$.
1: Initialize $\boldsymbol{w}^{(0)} = \boldsymbol{w}$;
2: **for** $t = 0, \ldots, T$ **do**
3:     Draw noise vectors $\boldsymbol{z}_1, \ldots, \boldsymbol{z}_m \sim \mathcal{N}(0, \sigma^2 I_{d\times d})$;
4:     Compute $\boldsymbol{g}_t = \frac{1}{m}\sum_{i=1}^m \nabla f(\boldsymbol{w}^{(t)} + \boldsymbol{z}_i, \boldsymbol{x})$;
5:     Update $\boldsymbol{w}^{(t+1)} \leftarrow (1 - \frac{\gamma}{\rho})\boldsymbol{w}^{(t)} - \gamma(\boldsymbol{g}_t - \frac{1}{\rho}\boldsymbol{w})$;
6:     **if** GroupSparse **then**
7:         Update $\boldsymbol{w}^{(t+1)} \leftarrow \mathrm{GST}_{\gamma\eta}(\boldsymbol{w}^{(t+1)} - \boldsymbol{w}) + \boldsymbol{w}$;
8:     **end if**
9: **end for**
10: Compute importance score $I(\boldsymbol{w}) = \left\|\frac{1}{\rho}(\boldsymbol{w}^{(T)} - \boldsymbol{w})\boldsymbol{w}\right\|_1$;
11: Prune network $f(\boldsymbol{w})$ according to $I(\boldsymbol{w})$;
12: Finetune pruned network w/ LoRA;
13: **Return** finetuned network;

---

### 4.2 LEVERAGING MOREAUGRAD FOR ROBUST PRUNING

**MoreauPruner**. With the defined MoreauGrad $\mathrm{MG}^\rho[g](\boldsymbol{w})$, we established a robust estimation on the influence of removing a weight element $\boldsymbol{w}^{(k)}$ based on Equation (3),

$$\mathrm{MG}^\rho\text{-}I(\boldsymbol{w}^{(k)}) = \|\mathrm{MG}^\rho[g](\boldsymbol{w}) \odot \boldsymbol{w}\|_1^{(k)}, \tag{9}$$

where $g(\boldsymbol{w})$ is the exception of loss function $L(\boldsymbol{w}, \mathcal{D})$ defined in Equation (1), and $\odot$ denotes Hadamard product.

We should note that the difference $\tilde{\boldsymbol{w}}_\rho^*(\boldsymbol{w}) - \boldsymbol{w}$ is aligned with $g^\rho$'s gradient (Moreau, 1965; Zhang & Farnia, 2023),

$$\nabla g^\rho(\boldsymbol{w}) = -\frac{1}{\rho}(\tilde{\boldsymbol{w}}_\rho^*(\boldsymbol{w}) - \boldsymbol{w}), \tag{10}$$

where the optimal solution $\tilde{\boldsymbol{w}}_\rho^*(\boldsymbol{w})$ of the optimization problem in Equation (6) can be obtained over a calibration dataset with the first-order gradient descent optimization method. By combining Equations (7) and (10), Equation (9) can be computed. Since Equation (9) is a robust version of Equation (3), the robust importance score of the structure $\boldsymbol{w}_i$ and group $\mathcal{G}$ can also be estimated by aggregating the importance score of each element with Equation (4) and Equation (5). We denote the structural pruning method removing groups with smaller robust importance score as MoreauPruner.

**MoreauPruner-GS**. Similar to MoreauPruner, the group-sparse robust estimation on the influence of removing parameter $\boldsymbol{w}^{(k)}$ is,

$$h\text{-MG}^\rho\text{-}I(\boldsymbol{w}^{(k)}) = \|h\text{-MG}^\rho[g_\sigma](\boldsymbol{w}) \odot \boldsymbol{w}\|_1^{(k)}, \tag{11}$$

where the group sparsity is conducted at the channel level in our implementation, *i.e.,* each variable subset in the 2, 1-group-norm is a channel in the model. To compute the GroupSparse-MoreauGrad $h\text{-MG}^\rho[g_\sigma](\boldsymbol{w})$, we utilize the proximal gradient descent algorithm as described in Algorithm 1. Note that we apply the group-soft-thresholding function as the proximal operator for the $\ell_{2,1}$-norm function present in GroupSparse-MoreauGrad,

$$\text{GST}_\alpha(\boldsymbol{v})_{S_i} := \begin{cases} 0 & \text{if } \|\boldsymbol{v}_{S_i}\|_2 \leq \alpha \\ \left(1 - \frac{\alpha}{\|\boldsymbol{v}_{S_i}\|_2}\right) \boldsymbol{v}_{S_i} & \text{if } \|\boldsymbol{v}_{S_i}\|_2 > \alpha. \end{cases}$$

Once the optimization of $h$-Moreau envelope ends, the GroupSparse-MoreauGrad $h\text{-MG}^\rho[g_\sigma](\boldsymbol{w})$ can be calculated according to Definition 3. We treat the method as MoreauPruner-GS to mark the group sparsity of importance score obtained during optimization.

After the pruning phase, a post-training with LoRA(Hu et al., 2021) is applied to the pruned model to recover model performance, as suggested by previous works.

## 5 NUMERICAL RESULTS

In this section, we conducted experiments on several famous LLMs to evaluate the proposed MoreauPruner's performance and support our theoretical claim. We also provided further insights in the discussion subsection on how and why MoreauPruner works well.

### 5.1 EXPERIMENTAL SETTINGS

**Pre-trained Models**. To demonstrate the versatility of MoreauPruner across different scales, we evaluate it on four open-source large language models: LLaMA-7B(Touvron et al., 2023), LLaMA-13B(Touvron et al., 2023), Vicuna-7B(Chiang et al., 2023) and LLaMA3-8B(AI@Meta, 2024).

**Evaluation**. Building on prior research (Zhang et al., 2023; Ma et al., 2023; Sun et al., 2023), we assess our method using seven zero-shot classification tasks on datasets centered around common sense reasoning: BoolQ (Clark et al., 2019), PIQA (Bisk et al., 2020), HellaSwag (Zellers et al., 2019), WinoGrande (Sakaguchi et al., 2021), ARC-easy (Clark et al., 2018), ARC-challenge (Clark et al., 2018), and OpenbookQA (Mihaylov et al., 2018). Consistent with (Gao et al., 2023), the model ranks options in multiple-choice tasks or generates answers for open-ended questions. Furthermore, we perform a zero-shot perplexity (PPL) analysis on WikiText2 (Merity et al., 2016) and PTB (Marcus et al., 1993) with 128-token segments, aligning our methodology with that of Zhang et al. (2023); Ma et al. (2023).

**Implementation Details**. In pruning phase, to align with the protocols of the closely related gradient-based method (Ma et al., 2023), our model pruning utilizes a calibration set of ten randomly selected, 128-token truncated sentences from the Bookcorpus (Zhu et al., 2015). The gradient of the Moreau envelope is computed using this calibration set, with the optimization step length fixed at ten. The pruning process typically completes in approximately 30 minutes on CPUs. In the post-training phase,

Table 1: Algorithms' robustness against weight perturbation. **Diff** denotes the absolute difference between weight formats, bfloat16 and float16. Rounding results in changes to the last digit.

(a) 0-shot PPL on WikiText2

| Method | Format | Pruning Ratio | | | |
|---|---|---|---|---|---|
| | | 5% | 10% | 15% | 20% |
| LLM-Pruner (Ma et al., 2023) | BF16 | 13.80 | 17.73 | 32.60 | 95.82 |
| | FP16 | 13.75 | 17.79 | 32.10 | 96.57 |
| | Diff(↓) | **0.05** | 0.07 | 0.51 | 0.75 |
| MoreauPruner | BF16 | 13.89 | 17.42 | 31.05 | 91.79 |
| | FP16 | 13.83 | 17.45 | 30.99 | 91.79 |
| | Diff(↓) | **0.05** | **0.03** | **0.06** | **0.00** |

(b) 0-shot PPL on PTB

| Method | Format | Pruning Ratio | | | |
|---|---|---|---|---|---|
| | | 5% | 10% | 15% | 20% |
| LLM-Pruner (Ma et al., 2023) | BF16 | 25.00 | 32.10 | 61.87 | 202.86 |
| | FP16 | 24.85 | 32.16 | 61.15 | 210.12 |
| | Diff(↓) | 0.15 | **0.06** | 0.72 | 7.26 |
| MoreauPruner | BF16 | 25.00 | 32.22 | 60.43 | 176.24 |
| | FP16 | 24.95 | 32.29 | 60.43 | 174.19 |
| | Diff(↓) | **0.05** | **0.06** | **0.00** | 2.05 |

we finetune the pruned model with a LoRA(Hu et al., 2021). A refined version of the Alpaca(Taori et al., 2023) dataset comprising about 50,000 samples is employed, with training extending over two epochs and generally taking three hours on a single NVIDIA RTX 3090 Ti GPU for 7B models. Detailed hyper-parameter selections are available in the appendices.

**Structural Pruning Baselines**. We compare MoreauPruner against two fundamental pruning techniques: *Magnitude* and *Random*. Magnitude pruning evaluates weight significance based on the magnitude of the weight matrix, whereas Random pruning indiscriminately removes weights. Additionally, we benchmark against three advanced alternatives: *LLM-Pruner* (Ma et al., 2023), which uses a gradient-based metric to determine weight importance; *LoraPrune* (Zhang et al., 2023), which utilizes a LoRA(Hu et al., 2021)-guided pruning criterion; and *WANDA* (Sun et al., 2023), designed for unstructured or semi-structured pruning but adaptable to other structural frameworks.

We also introduce *SmoothGrad*, a preliminary version of MoreauPruner that enhances network smoothness by applying Gaussian smoothing during the inference, as we explained in Theorem 1. The importance scores are estimated with the smoothed gradient using Equation (3), and the we still remove those parameter groups with lower importance scores. A thorough comparison of these methods is documented in appendices.

## 5.2 ROBUSTNESS AGAINST WEIGHT PERTURBATION

As we previously discussed, few-shot gradient-based pruning methods are significantly influenced by the changes of the gradient. Even minor differences on model weight can lead to markedly different pruning outcomes. In contrast, MoreauGrad is theoretically robust against norm-bounded weight perturbation. To validate this assertion, we adhered to the channel-wise pruning protocol established in prior research (Ma et al., 2023), removing a fixed ratio of channels based on their importance score.

When utilizing 16 bits to store a float number, BF16 has larger range while FP16 has better precision. Considering that some LLMs are trained on BF16 and inferred on FP16, our experiments were conducted on the LLaMA-7B model using both BF16 and FP16 weight bit formats. We standardized the calibration sample selection during pruning across different settings to ensure a fair comparison. Upon completing the pruning process, we performed a zero-shot perplexity (PPL) analysis using 128-token segments on the WikiText2 and PTB datasets and compared the discrepancies between FP16 and BF16. The findings are presented in Table 1. Due to the space limitation, we put the full evaluation results on zero-shot question-answering in appendices.

The results indicate that, for MoreauPruner, the performance under different weight format is closer to each other, which indicates a better consistency of pruning outcomes. The result demonstrates the robustness of MoreauPruner against weight perturbations caused by different weight formats.

## 5.3 COMPARISON ON CHANNEL IMPORTANCE ESTIMATION

As detailed in previous sections (Section 3.2 and Section 4.2), both MoreauPruner and certain existing structured pruning algorithms assign an importance score to each channel within a given layer. Channels deemed less important are more likely to be pruned. To better understand the functionality of MoreauPruner, we conducted a detailed comparison of channel importance estimation.

Within each weight structure (e.g., a layer) of the model, all channels are ranked according to the importance scores assigned by different pruning algorithms. A higher ranking indicates a more

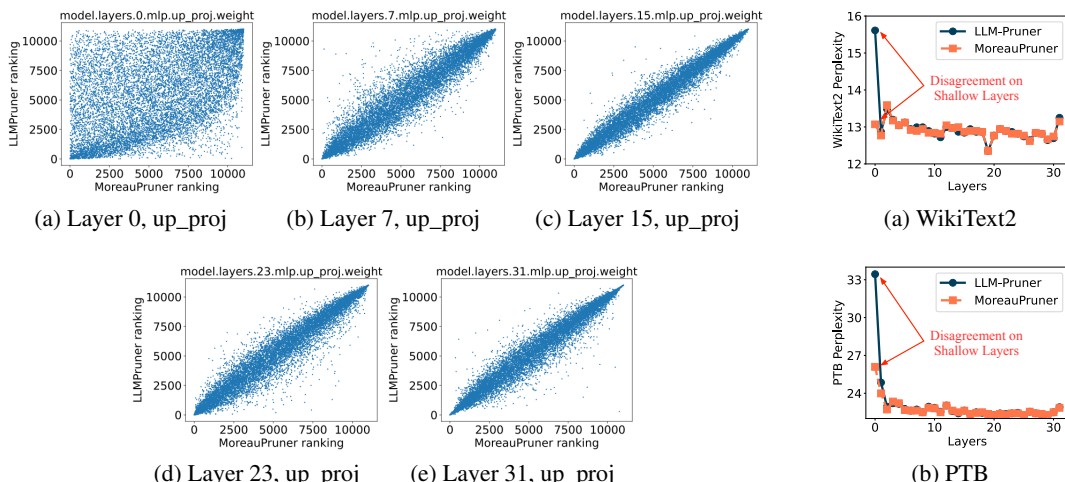

Figure 2: The channel ranking in up projector module of LLaMA-7B given by our method and baseline algorithm. A large ranking denotes a more important channel.

Figure 3: Only pruning weight in a single layer.

important channel. We compared the channel rankings generated by our main competitor, LLM-Pruner (Ma et al., 2023), and MoreauPruner. The ranking pairs are plotted in a figure to illustrate the (in)consistency between the two methods. Some examples from up projector module from different layers of LLaMA-7B are shown in Figure 2. The ranking relationship indicates that the significant disagreement between MoreauPruner and gradient-based baseline occurs on shallow layers (closer to input), where the gradient is known as fragile due to the accumulation during the long backward propagation process.

We conducted an experiment in Figure 3 to investigate the effect of the disagreement. For each time, we remove the weight selected by algorithms in a single layer and test the zero-shot perplexity on both datasets. The significant performance gap occurs on shallow layers between MoreauPruner and gradient-based baseline, which shows the effectiveness of our method in distinguishing essential weights. Extended examples and detailed analyses can be found in the Appendices.

### 5.4  ZERO-SHOT PERFORMANCE

We have developed two variants of our method, named *MoreauPruner* and *MoreauPruner-GS*, according to whether the sparsity penalty is applied. These techniques were tested on four pretrained models: LLaMA-7B, Vicuna-7B, LLaMA-13B and LLaMA3-8B, with their performance detailed in Tables 2 to 5 and we also provide full evaluation results in appendices. We should note that the pruning setting in the literature varies among different works. We keep the same experiment setting with our primary baseline LLM-Pruner for a fair comparison. LoRAPrune and WANDA (marked as ‡) do not share the same protocols with our main experiments as we detailed in appendices; we list the results here for reference.

The evaluations indicate that MoreauPruner can effectively maintain the model performance. For example, for the largest model LLaMA-13B, We have noticed that the performance gap between the compressed and original models is closer than that of the 7B models. After a quick recovery, the zero-shot accuracy of the compressed with 80% parameters is nearly equivalent to the original model's performance (64.94% vs. 64.97%). Such a phenomenon may indicate more redundant weights in the larger models. In other words, those huge LLMs ($\geq$13B) can be potentially inferred with less trade-off on performance. On other models, we have observed that with a 20% reduction in parameters on LLaMA-7B, *MoreauPruner* and *MoreauPruner-GS* maintains 95.48% and 95.64% of the original performance with a quick post-training. On Vicuna-7B, *MoreauPruner-GS* maintains 96.16% of the original performance. An interesting fact is that on Vicuna-7B and LLaMA-13B, *MoreauPruner-GS* surpassed *MoreauPruner* by a notable margin, thanks to the structural sparsity introduced during optimization.

Table 2: Zero-shot performance of the compressed LLaMA-13B. $*$ is implemented according to open-source code. The **best** results is bold. The methods proposed in this paper are filled with green.

| Pruning Ratio | Method | WikiText2(↓) | PTB(↓) | BoolQ | PIQA | HellaSwag | WinoGrande | ARC-e | ARC-c | OBQA | QA-Average |
|---|---|---|---|---|---|---|---|---|---|---|---|
| Ratio = 0% | LLaMA-13B† | 11.58 | 44.54 | 68.47 | 78.89 | 76.24 | 70.09 | 74.58 | 44.54 | 42.00 | 64.97 |
| Ratio = 20% w/o finetune | LLM-Pruner*(Ma et al., 2023) | **16.43** | **59.96** | 63.00 | 77.53 | 73.79 | 64.33 | 69.07 | 40.96 | 40.60 | 61.33 |
| | SmoothGrad | 16.55 | **59.96** | 62.94 | 77.04 | 73.78 | 65.98 | 68.35 | 40.53 | 41.40 | 61.43 |
| | MoreauPruner | 16.95 | 61.39 | 62.48 | 77.64 | 73.61 | 66.38 | 67.47 | 39.68 | 40.60 | 61.12 |
| | MoreauPruner-GS | 17.11 | 61.39 | 72.97 | 77.53 | 74.44 | 64.09 | 66.08 | 40.44 | 41.40 | **62.42** |
| Ratio = 20% w/ finetune | LLM-Pruner*(Ma et al., 2023) | 15.04 | 57.00 | 67.28 | 79.00 | 75.13 | 69.06 | 71.68 | 41.89 | 43.60 | 63.95 |
| | SmoothGrad | **15.01** | **56.55** | 66.39 | 79.05 | 74.95 | 69.46 | 71.17 | 42.75 | 43.40 | 63.88 |
| | MoreauPruner | 15.52 | 57.44 | 64.86 | 79.22 | 75.07 | 70.48 | 71.68 | 43.60 | 42.80 | 63.96 |
| | MoreauPruner-GS | 15.28 | 57.67 | 75.17 | 78.24 | 74.77 | 68.19 | 70.12 | 43.09 | 45.00 | **64.94** |

Table 3: Zero-shot performance of the Pruned LLaMA-7B. † denotes results from Ma et al. (2023) and ‡ denotes results from Zhang et al. (2023).

| Pruning Ratio | Method | WikiText2(↓) | PTB(↓) | BoolQ | PIQA | HellaSwag | WinoGrande | ARC-e | ARC-c | OBQA | QA-Average |
|---|---|---|---|---|---|---|---|---|---|---|---|
| Ratio = 0% | LLaMA-7B† | 12.62 | 22.14 | 73.18 | 78.35 | 72.99 | 67.01 | 67.45 | 41.38 | 42.40 | 63.25 |
| Ratio = 20% w/o finetune | Magnitude† | 582.41 | 1022.17 | 59.66 | 58.00 | 37.04 | 52.41 | 33.12 | 28.58 | 29.80 | 42.65 |
| | Random† | 27.51 | 43.19 | 61.83 | 71.33 | 56.26 | 54.46 | 57.07 | 32.85 | 35.00 | 52.69 |
| | WANDA‡(Sun et al., 2023) | 22.12 | 38.19 | 64.93 | 70.14 | 58.12 | 55.39 | 56.63 | 33.98 | 35.43 | 53.23 |
| | LLM-Pruner*(Ma et al., 2023) | 19.09 | 34.23 | 56.91 | 75.08 | 66.81 | 60.06 | 60.94 | 36.43 | 40.00 | 56.60 |
| | LoRAPrune‡(Zhang et al., 2023) | 20.67 | 34.12 | 57.98 | 75.11 | 65.81 | 59.90 | 62.14 | 34.59 | 39.98 | 56.50 |
| | SmoothGrad | 18.91 | 34.30 | 59.60 | 75.14 | 65.98 | 61.01 | 60.77 | 37.12 | 39.80 | 57.06 |
| | MoreauPruner | **18.61** | **32.92** | 55.44 | 76.17 | 66.47 | 63.61 | 61.53 | 37.80 | 40.60 | **57.37** |
| | MoreauPruner-GS | 18.72 | 34.91 | 62.51 | 75.52 | 68.29 | 62.75 | 54.88 | 36.35 | 40.80 | 57.30 |
| Ratio = 20% w/ finetune | WANDA‡(Sun et al., 2023) | 18.43 | 33.16 | 65.75 | 74.70 | 64.52 | 59.35 | 60.65 | 36.26 | 39.40 | 57.23 |
| | LLM-Pruner*(Ma et al., 2023) | 17.62 | 30.57 | 65.78 | 76.44 | 68.67 | 64.33 | 63.26 | 36.35 | 41.00 | 59.40 |
| | LoRAPrune‡(Zhang et al., 2023) | 16.80 | **28.75** | 65.62 | 79.31 | 70.00 | 62.76 | 65.87 | 37.69 | 39.14 | 60.05 |
| | SmoothGrad | 17.45 | 30.57 | 66.48 | 76.99 | 68.64 | 65.35 | 63.68 | 37.80 | 41.00 | 59.99 |
| | MoreauPruner | 17.01 | 30.27 | 66.61 | 77.04 | 68.32 | 65.59 | 65.57 | 38.40 | 41.20 | 60.39 |
| | MoreauPruner-GS | **16.65** | 30.69 | 68.87 | 77.26 | 69.81 | 65.04 | 63.64 | 38.23 | 40.60 | **60.49** |

Table 4: Zero-shot performance of the compressed Vicuna-7B. Full results can be found in Appendices.

| Pruning Ratio | Method | WikiText2(↓) | PTB(↓) | QA-Average |
|---|---|---|---|---|
| Ratio = 0% | Vicuna-7B† | 16.11 | 61.39 | 62.71 |
| Ratio = 20% w/o finetune | Magnitude† | 3539.98 | 5882.21 | 40.41 |
| | Random† | 34.63 | 112.44 | 52.18 |
| | LLM-Pruner* | 25.74 | **92.87** | 56.18 |
| | SmoothGrad | 25.99 | **92.87** | 56.17 |
| | MoreauPruner | **25.54** | 94.34 | **56.76** |
| | MoreauPruner-GS | 30.69 | 108.16 | **56.76** |
| Ratio = 20% w/ finetune | LLM-Pruner* | 19.47 | 72.33 | 57.72 |
| | SmoothGrad | 19.51 | **72.05** | 57.64 |
| | MoreauPruner | 19.66 | 73.47 | 58.60 |
| | MoreauPruner-GS | **19.13** | 73.76 | **60.03** |

Table 5: Results on LLaMA3-8B.

| Pruning Ratio | Method | WikiText2(↓) | PTB(↓) | QA-Average |
|---|---|---|---|---|
| Ratio = 0% | LLaMA-8B† | 14.14 | 27.98 | 70.33 |
| Ratio = 20% w/o finetune | LLM-Pruner | 25.74 | 45.69 | 58.29 |
| | MoreauPruner-GS | **25.40** | **43.78** | **60.68** |
| Ratio = 20% w/ finetune | LLM-Pruner | 23.71 | 42.01 | 64.11 |
| | MoreauPruner-GS | **22.98** | **39.25** | **65.37** |

Table 6: Larger recovery set boosts performance.

| Pruning Ratio | Method | Recovery Set | QA-Average |
|---|---|---|---|
| Ratio = 0% | LLaMA-7B† | N/A | 63.25 |
| Ratio = 20% w/ finetune | MoreauPruner-GS | Alpaca(50k) | 60.49 |
| | MoreauPruner-GS | LaMini(2.59M) | 63.17 (+2.68) |

According the the results on strongest foundational model LLaMA3-8B that is pre-trained with more high-quality data compared with previous version. *MoreauPruner-GS* still works well without any hyper-parameter modification. We have noticed that the performance of pruned LLaMA3-8B drops more than that of LLaMA-7B. This may lead by the fact that the pretraining of LLaMA3-8B is more sufficient according to official report and there is less redundant model weight. However, the pruned LLaMA3-8B still beats the original LLaMA-7B by a noticeable margin (63.25% vs. 65.37%).

## 5.5 FURTHER DISCUSSION

In this subsection, we extended our experiment to identify how *MoreauPruner* works. We also discussed that with more computational resource, how can *MoreauPruner* be further improved.

**Effect of Function Smoothing**. In our preliminary evaluations, we introduced *SmoothGrad* to assess the impact of function smoothing. This approach often matches or exceeds the performance of gradient-based competitors. Notably, on the benchmark model LLaMA-7B, *SmoothGrad* outperformed all baseline methods prior to finetuning. These findings suggest that gradient-based pruning methods could benefit from function smoothing, as it helps mitigate the excessive sharpness of certain parameters within the differential space.

Table 7: The performance of the MoreauPruner-GS on LLaMA-7B with different calibration set size. We repeat three times and report the mean and variance for each setting.

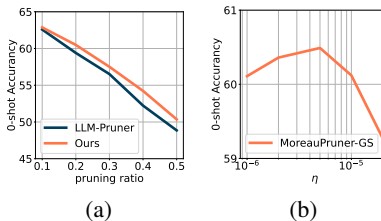

(a)   (b)

Figure 4: The effect of different (a) pruning ratio; (b) hyper-paramter $\eta$.

| Pruning Ratio | Calibration | WikiText2($\downarrow$) | PTB($\downarrow$) | QA-Average |
|---|---|---|---|---|
| Ratio = 20% w/o finetune | 10 | 18.72±0.29 | 34.91±0.85 | 57.30±0.53 |
| | 1000 | **18.50±0.10** | **32.16±0.19** | **58.12±0.31** |
| Ratio = 20% w/ finetune | 10 | **16.65±0.16** | 30.69±**0.21** | 60.49±0.26 |
| | 1000 | 16.95±**0.09** | **30.21**±0.23 | **60.63±0.12** |

**Larger Recovery Set**. In the main experiments, the recovery phase was conducted on Alpaca(Taori et al., 2023), utilizing a dataset of 50k samples. To demonstrate the potential enhancement achieved by the pruned model, we carried out an experiment on a significantly larger dataset, LaMini(Wu et al., 2023), consisting of 2.59 million samples. The findings, presented in Table 6, reveal that the performance of the compressed model closely approximates that of the base model (63.17% v.s. 63.25%), respectively. These results further substantiate the hypothesis of the presence of redundant weights in LLMs.

**Larger Calibration Set & Randomness Analyses**. To be strictly aligned with our primary baseline and have a fair comparison, we utilize only ten randomly picked samples as the calibration set to judge the importance score of weight. Unavoidably, the small calibration set introduces randomness to model performance. To further evaluate our method, we enlarge the size of the calibration set utilized during the pruning phase. We found that a larger calibration set can efficiently improve pruning quality and reduce randomness in performance as shown in Table 7. Estimating gradient importance on 1000 samples raises the average zero-shot accuracy from 57.30% to 58.12% and decreases PPL by 0.22 and 2.75 on WikiText2 and PTB. However, the difference in post-finetuning performance is shrinking, resulting in only a 0.14% difference in average accuracy.

**Effect of Pruning Ratio**. We explored the influence of varying pruning ratios as illustrated in Figure 4a. It is evident that our methods consistently work well across different pruning ratios. This stability underscores the robustness and effectiveness of our pruning strategies.

**Impact of Hyper-parameters**. The hyper-parameter $\eta$ controls the ratio of group-sparsity of MoreauPruner-GS during optimization. We conduct an ablation study on LLaMA-7B with 20% sparsity to evaluate the impact of different hyper-parameter values $\eta$. The results illustrated in Figure 4b give the average 0-shot accuracy after finetuning. According to the results, we choose $\eta$=5e-6 for all the experiments in this paper.

## 6 CONCLUSION

In this paper, we discussed how minor changes in model weights can lead to unstable pruning results for large language models (LLMs). To address this instability, we introduced MoreauPruner, a weight-perturbation structural pruning method. Our theoretical analysis demonstrates that MoreauPruner is robust to norm-bounded perturbations. Numerical experiments conducted on well-known LLMs suggest that MoreauPruner can efficiently compress LLMs while maintaining their performance. For future work, we propose combining structural pruning technology with other model compression methods to accelerate model inference and reduce computational costs.

**Limitations**. The authors acknowledge that the number of parameters utilized in the models for this paper only reach 13B due to limited hardware budget. The performance of MoreauPruner on extremely large-scale models (e.g., 30B, 70B, etc.) will be further explored once enough hardware resources are available.

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

# A PROOF

## A.1 PROOF OF THEOREM 1

As Theorem 1 assumes a Lipschiz function $g$, we can apply the Stein's lemma (Landsman et al., 2013) to show

$$\nabla g_\sigma(\boldsymbol{w}) = \mathbb{E}[\nabla g(\boldsymbol{w} + \mathbf{Z})] = \mathbb{E}[g(\boldsymbol{w} + \mathbf{Z})\frac{\mathbf{Z}}{\sigma^2}]$$

Therefore, for every $\boldsymbol{w}, \boldsymbol{w}' \in \mathbb{R}^d$ and unit-$\ell_2$-norm vector $\|\boldsymbol{u}\|_2 = 1$ we have the following

$$
\begin{aligned}
\left|\boldsymbol{u}^\top (\nabla g_\sigma(\boldsymbol{w}) - \nabla g_\sigma(\boldsymbol{w}'))\right| &= \left|\boldsymbol{u}^\top (\mathbb{E}[\nabla g(\boldsymbol{w} + \mathbf{Z})] - \mathbb{E}[\nabla g(\boldsymbol{w}' + \mathbf{Z})])\right| \\
&= \left|\boldsymbol{u}^\top (\mathbb{E}\left[\frac{\mathbf{Z}}{\sigma^2}g(\boldsymbol{w} + \mathbf{Z})\right] - \mathbb{E}\left[\frac{\mathbf{Z}}{\sigma^2}g(\boldsymbol{w}' + \mathbf{Z})\right])\right| \\
&= \left|\mathbb{E}\left[\frac{\boldsymbol{u}^\top \mathbf{Z}}{\sigma^2}(g(\boldsymbol{w} + \mathbf{Z}) - g(\boldsymbol{w}' + \mathbf{Z}))\right]\right| \\
&\le \mathbb{E}\left[\frac{|\boldsymbol{u}^\top \mathbf{Z}|}{\sigma^2}|g(\boldsymbol{w} + \mathbf{Z}) - g(\boldsymbol{w}' + \mathbf{Z})|\right] \\
&\le \mathbb{E}\left[\frac{|\boldsymbol{u}^\top \mathbf{Z}|}{\sigma^2}\beta\|\boldsymbol{w} - \boldsymbol{w}'\|_2\right] \\
&= \frac{\beta\|\boldsymbol{w} - \boldsymbol{w}'\|_2}{\sigma}\mathbb{E}\left[|\frac{\boldsymbol{u}^\top \mathbf{Z}}{\sigma}|\right] \\
&\le \frac{\beta\|\boldsymbol{w} - \boldsymbol{w}'\|_2}{\sigma}.
\end{aligned}
$$

In the above, note that $\frac{\mathbf{u}^\top \mathbf{Z}}{\sigma} \sim \mathcal{N}(0, 1)$. As a result, the gradient of $g_\sigma$ will be $\frac{\beta}{\sigma}$-Lipschitz, and $g_\sigma$ is $\frac{\beta}{\sigma}$-smooth, which means for every $\boldsymbol{w}, \boldsymbol{w}'$ we have,

$$\left|g_\sigma(\boldsymbol{w}') - \nabla g_\sigma(\boldsymbol{w})^\top (\boldsymbol{w}' - \boldsymbol{w})\right| \le \frac{\beta}{2\sigma}\|\boldsymbol{w} - \boldsymbol{w}'\|_2^2$$

As a result, $\Theta(\boldsymbol{w}) = g_\sigma(\boldsymbol{w}) + \frac{\beta}{2\sigma}\|\boldsymbol{w}\|_2^2$ will be a convex function. Therefore, we can rewrite the definition of the Moreau envelope as

$$
\begin{aligned}
g_\sigma^\rho(\boldsymbol{w}) &= \min_{\tilde{\boldsymbol{w}} \in \mathbb{R}^d} \Theta(\tilde{\boldsymbol{w}}) - \frac{\beta}{2\sigma}\|\tilde{\boldsymbol{w}}\|_2^2 + \frac{1}{2\rho}\|\tilde{\boldsymbol{w}} - \boldsymbol{w}\|_2^2 \\
&= \min_{\tilde{\boldsymbol{w}} \in \mathbb{R}^d} \Theta(\tilde{\boldsymbol{w}}) + (\frac{1}{2\rho} - \frac{\beta}{2\sigma})\|\tilde{\boldsymbol{w}}\|_2^2 - \frac{1}{\rho}\boldsymbol{w}^\top \tilde{\boldsymbol{w}} + \frac{1}{2\rho}\|\boldsymbol{w}\|_2^2 \\
&= \frac{1}{2\rho}\|\boldsymbol{w}\|_2^2 - \frac{1}{\rho}\max_{\tilde{\boldsymbol{w}} \in \mathbb{R}^d}\left\{\boldsymbol{w}^\top \tilde{\boldsymbol{w}} - \rho\Theta(\tilde{\boldsymbol{w}}) - \frac{\sigma - \rho\beta}{2\sigma}\|\tilde{\boldsymbol{w}}\|_2^2\right\}.
\end{aligned}
$$

Therefore, $\rho g_\sigma^\rho(\boldsymbol{w})$ is the subtraction of the Fenchel conjugate of $\boldsymbol{c}(\boldsymbol{w}) = \rho\Theta(\boldsymbol{w}) + \frac{\sigma - \rho\beta}{2\sigma}\|\tilde{\boldsymbol{w}}\|_2^2$ from the 1-strongly-convex $\frac{1}{2}\|\boldsymbol{w}\|_2^2$. Then, we apply the result that the Fenchel conjugate of a $\mu$-strongly convex function is $\frac{1}{\mu}$-smooth convex function in Zhou (2018). Therefore, the following Fenchel conjugate

$$\boldsymbol{c}^*(\boldsymbol{w}) := \max_{\tilde{\boldsymbol{w}} \in \mathbb{R}^d}\left\{\boldsymbol{w}^\top \tilde{\boldsymbol{w}} - \rho\Theta(\tilde{\boldsymbol{w}}) - \frac{\sigma - \rho\beta}{2\sigma}\|\tilde{\boldsymbol{w}}\|_2^2\right\}$$

is a $\frac{\sigma}{\sigma - \rho\beta}$-smooth convex function. Since, we subtract two convex functions from each other where the second one has a constant Hessian $I$, then the resulting function will be smooth of the following degree:

$$\frac{1}{\rho}\max\left\{|\frac{\sigma}{\sigma - \rho\beta} - 1|, |0 - 1|\right\} = \frac{\sigma}{\min\{\sigma\rho, \sigma - \rho\beta\}},$$

which completes the proof of the theorem.

## A.2 PROOF OF THEOREM 2

To prove Theorem 2, we note that the additional $h$ is a convex function. Given the formulation of the $h$-Moreau envelope of $g_\sigma^\rho(\boldsymbol{w})$ and the assumption $0 < \rho < \frac{\sigma}{\beta}$ in the theorem, we have

$$g_{\sigma,h}^\rho(\boldsymbol{w}) := \min_{\tilde{\boldsymbol{w}} \in \mathbb{R}^d} g_\sigma(\tilde{\boldsymbol{w}}) + \frac{1}{2\rho}\|\tilde{\boldsymbol{w}} - \boldsymbol{w}\|_2^2 + h(\tilde{\boldsymbol{w}} - \boldsymbol{w}),$$

$$= \min_{\tilde{\boldsymbol{w}} \in \mathbb{R}^d} \Theta(\tilde{\boldsymbol{w}}) + (\frac{1}{2\rho} - \frac{\beta}{2\sigma})\|\tilde{\boldsymbol{w}}\|_2^2 - \frac{1}{\rho}\boldsymbol{w}^\top \tilde{\boldsymbol{w}} + \frac{1}{2\rho}\|\boldsymbol{w}\|_2^2 + h(\tilde{\boldsymbol{w}} - \boldsymbol{w}),$$

where $\Theta(\boldsymbol{w}) = g_\sigma(\boldsymbol{w}) + \frac{\beta}{2\sigma}\|\boldsymbol{w}\|_2^2$ is a convex function. Then the function $\phi : \mathbb{R}^d \to \mathbb{R}$ defined as

$$\phi(\tilde{\boldsymbol{w}}) = (\frac{1}{2\rho} - \frac{\beta}{2\sigma})\|\tilde{\boldsymbol{w}}\|_2^2 - \frac{1}{\rho}\boldsymbol{w}^\top \tilde{\boldsymbol{w}}$$

is a $\frac{\sigma - \rho\beta}{\sigma\rho}$-strongly-convex function. As a result, $\Theta(\tilde{\boldsymbol{w}}) + \phi(\tilde{\boldsymbol{w}}) + h(\tilde{\boldsymbol{w}} - \boldsymbol{w})$ is strongly-convex function with strong-convexity degree $\frac{\sigma - \rho\beta}{\sigma\rho}$. Therefore, the optimization of $h$-Moreau envelope has a unique locally and globally optimal solution. we define the proximal operator of $h$ function as

$$\text{prox}_{h(\cdot)}(\boldsymbol{w}) := \arg\min_{\boldsymbol{w}' \in \mathbb{R}^d} h(\boldsymbol{w}') + \frac{1}{2}\|\boldsymbol{w}' - \boldsymbol{w}\|_2^2.$$

Then since the objective function of $h$-Moreau envelope consists of the following two convex functions (w.r.t. $\boldsymbol{\delta} := \tilde{\boldsymbol{w}} - \boldsymbol{w})$ $t_{\boldsymbol{w}}(\boldsymbol{\delta}) := g_\sigma(\boldsymbol{w} + \boldsymbol{\delta}) + \frac{1}{2\rho}\|\boldsymbol{\delta}\|_2^2$ and $h(\boldsymbol{\delta})$, the optimal solution $\boldsymbol{\delta}^*$ will satisfy the following equation with $\gamma > 0$:

$$\boldsymbol{\delta}^* = \text{prox}_{\gamma h(\cdot)}\big(\boldsymbol{\delta}^* - \gamma\nabla t_{\boldsymbol{w}}(\boldsymbol{\delta}^*)\big) \overset{\gamma = \rho}{=} \text{prox}_{\rho h(\cdot)}\big(-\rho\nabla g_\sigma(\boldsymbol{w} + \boldsymbol{\delta}^*)\big).$$

The above implies that, if we use $\psi$ to denote the identity map we will get:

$$\boldsymbol{\delta}^*(\boldsymbol{w}) = \Big((\psi + \text{prox}_{\rho h(\cdot)} \circ \rho\nabla g_\sigma)^{-1} - \psi\Big)(\boldsymbol{w}).$$

Note that in the above $\psi + \text{prox}_{\rho h(\cdot)} \circ \rho\nabla g_\sigma$ will be a $(1 - \frac{\rho\beta}{\sigma})$-monotone operator, where we call $t : \mathbb{R}^d \to \mathbb{R}^d$ $\tau$-monotone if for every $\boldsymbol{w}, \boldsymbol{v} \in \mathbb{R}^d$:

$$(\boldsymbol{v} - \boldsymbol{w})^\top\big(t(\boldsymbol{v}) - t(\boldsymbol{w})\big) \geq \tau\|\boldsymbol{v} - \boldsymbol{w}\|_2^2.$$

The monotonicity arises because the gradient of a $\lambda$-weakly convex function is -$\lambda$-monotone, and the proximal operator is known to be 1-monotone. Hence, $\boldsymbol{\delta}^*(\boldsymbol{w})$ will be a Lipschitz function with the following Lipschitz constant (note that $(\psi + \text{prox}_{\rho h(\cdot)} \circ \rho\nabla g_\sigma)^{-1}$ is a monotone function with a degree between 0 and $\frac{\sigma}{\sigma - \rho\beta}$):

$$\max\left\{|\frac{\sigma}{\sigma - \rho\beta} - 1|, |0 - 1|\right\} = \max\left\{\frac{\rho\beta}{\sigma - \rho\beta}, 1\right\}.$$

Therefore, for any given convex function $h$, the $h$-MoreauGrad

$$h\text{-MG}^\rho[g](\mathbf{w}) := \frac{1}{\rho}\boldsymbol{\delta}^*(\boldsymbol{w})$$

will be a Lipschitz function with the constant $\frac{\sigma}{\min\{\sigma\rho, \sigma - \rho\beta\}}$. Then the proof the theorem is finished.

## B EXTENDED COMPARISON ON CHANNEL RANKING ASSIGNMENT

In Section 5.2 of the main text, we demonstrate the inconsistency in ranking assignments across different layer depth. Here, we extend the discussion by exploring how this inconsistency happens on different modules. Several examples from different layers are illustrated in Figures 5 and 6. The experimental results indicate that the major disagreements between MoreauPruner and gradient-based methods occur in the most shallow and deepest layers. Given that these layers are known to be sensitive to pruning (Ma et al., 2023; Yin et al., 2023; Ji et al., 2023), the performance gap between gradient-based methods and MoreauPruner can be partially attributed to differences in channel pruning within these layers. Additionally, we observed that the ranking stability among the middle layers suggests that the weights in these layers of LLMs may have converged to a flatter minimum, as both gradient-based and robust-gradient-based measurements yield similar sensitivity rankings. The numerical results also suggest that while gradient-based methods and MoreauPruner generally agree on the importance of channels within the attention module, there is more disagreement in the feed-forward network (FFN) module.

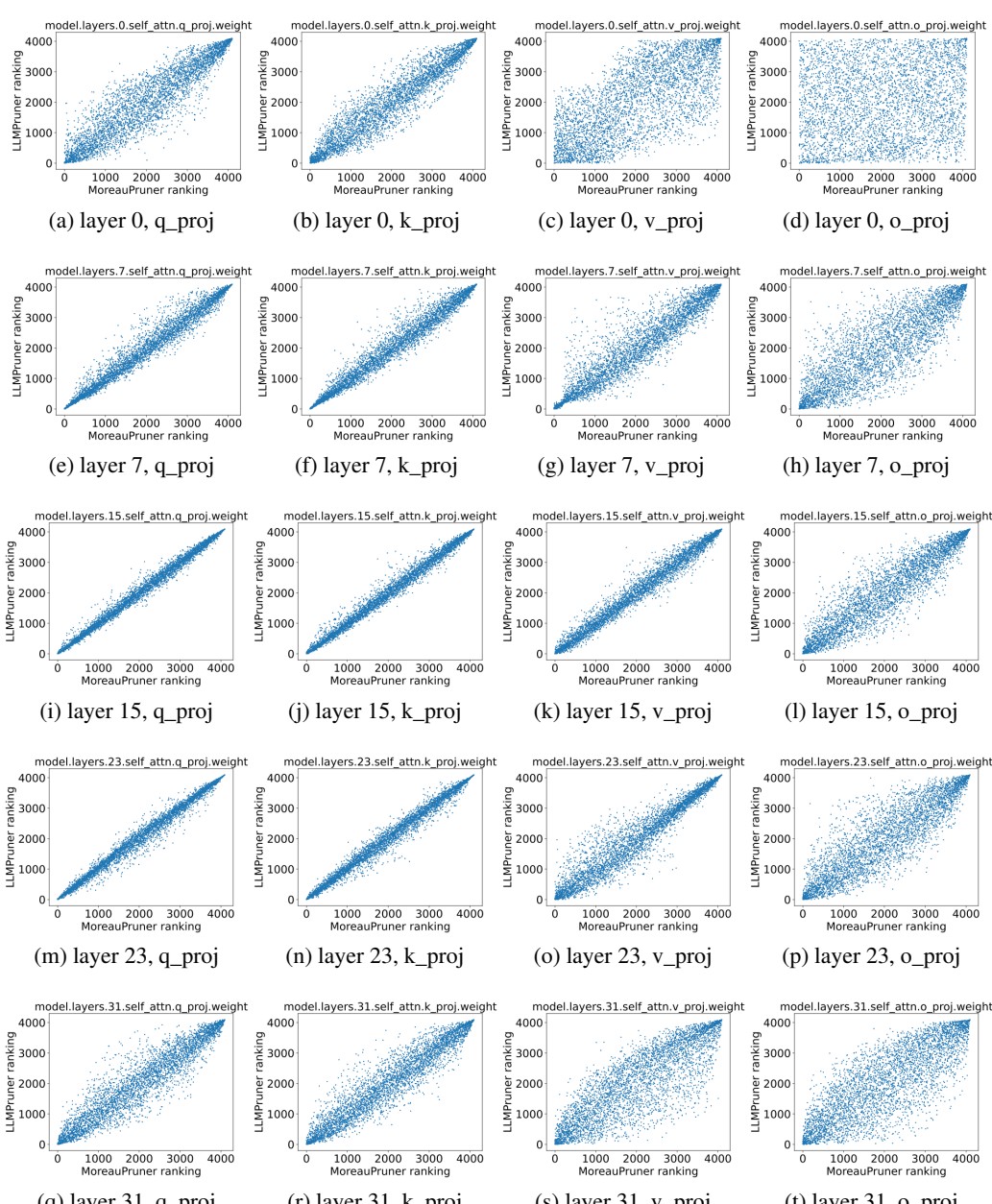

Figure 5: The channel ranking in attention module with different layer depths in LLaMA-7B given by different algorithms.

## C    COMPLETE RESULTS FOR TABLES IN THE MAIN TEXT

We include the full evaluation results on the effect of weight perturbation between BF16 and FP16 in Table 8. The column labeled **Diff** represents the difference between the BF16 and FP16 columns, indicating sensitivity to weight perturbation. Notably, MoreauPruner shows a lower difference in most cases, demonstrating the consistency of pruning results. Furthermore, MoreauPruner often yields better PPL (Perplexity) and QA (Question-Answering) accuracy, indicating superior performance.

Additionally, we provide the complete evaluation results for Vicuna-7B and LLaMA3-8B in Tables 9 and 10. The full results of the effect of larger recovery set are illustrated in Table 11.

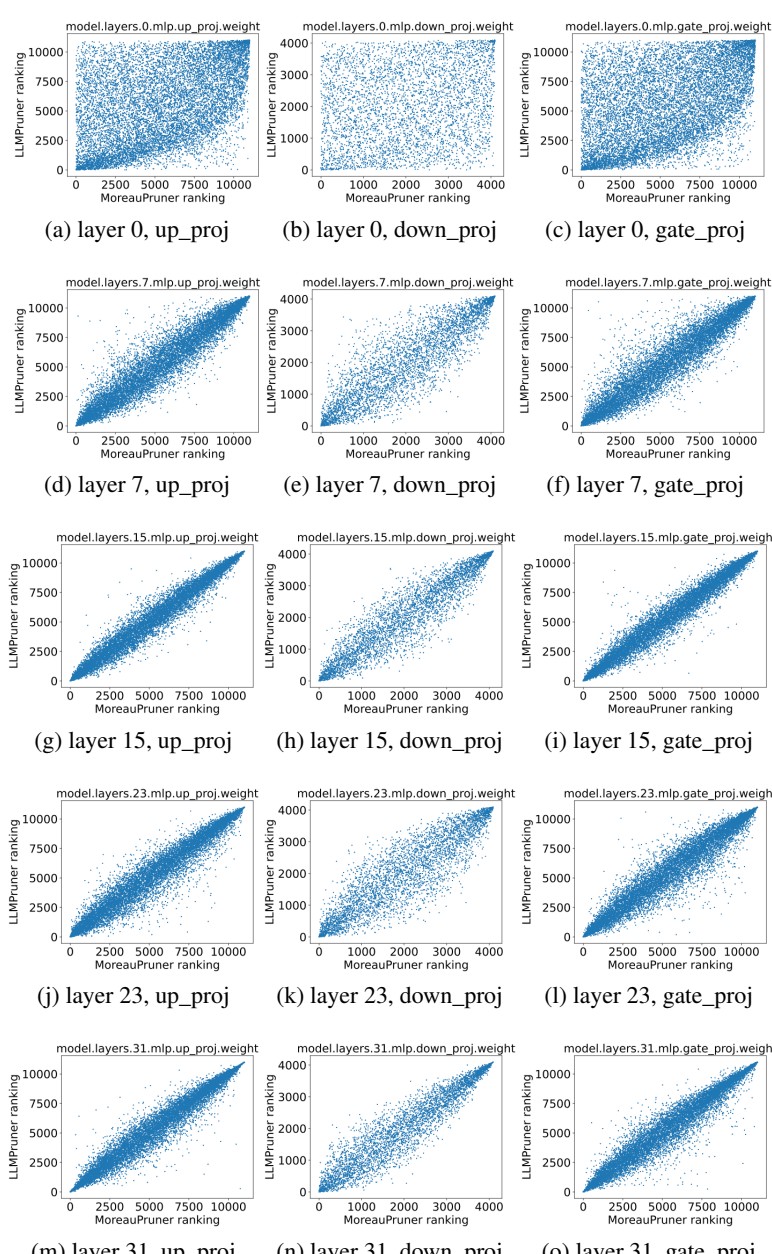

Figure 6: The channel ranking in feed-forward network module with different layer depths in LLaMA-7B given by different algorithms.

## D  COMPARED WITH SCRATCH TRAINING

We compare our MoreauPruner-GS with StableLM-3B[1] with a similar parameter size. With MoreauPruner-GS, We prune LLaMA-7B and get a compact model with 3.45B parameters. Both models are finetuned on Alpaca(Taori et al., 2023) dataset for a fair comparison. The result can be found in Table 12. MoreauPruner-GS sometimes achieves better results compared with LLMs that are trained from scratch. We also recognize that the pruned model may not consistently surpass other models with similar scale, due to the significant disparity in the size of the training corpus.

[1]https://huggingface.co/stabilityai/stablelm-tuned-alpha-3b

Table 8: Full evaluation results of weight perturbation on LLaMA-7B (w/o finetune).

| Pruning Ratio | Method | Format | WikiText2(↓) | PTB(↓) | BoolQ | PIQA | HellaSwag | WinoGrande | ARC-e | ARC-c | OBQA | QA-Average |
|---|---|---|---|---|---|---|---|---|---|---|---|---|
| 5% | LLM-Pruner (Ma et al., 2023) | BF16 | 13.80 | 25.00 | 71.83 | 76.39 | 69.77 | 66.14 | 63.93 | 38.99 | 40.40 | 61.06 |
| | | FP16 | 13.75 | 24.85 | 71.68 | 76.22 | 69.75 | 66.54 | 63.76 | 39.16 | 40.40 | 61.07 |
| | | Diff(↓) | **0.05** | 0.15 | - | - | - | - | - | - | - | **0.01** |
| | MoreauPruner | BF16 | 13.89 | 25.00 | 72.39 | 76.55 | 70.07 | 66.38 | 65.24 | 39.68 | 40.60 | 61.56 |
| | | FP16 | 13.83 | 24.95 | 72.17 | 76.33 | 70.17 | 66.93 | 65.28 | 39.25 | 40.40 | 61.50 |
| | | Diff(↓) | **0.05** | **0.05** | - | - | - | - | - | - | - | 0.05 |
| 10% | LLM-Pruner (Ma et al., 2023) | BF16 | 17.73 | 32.10 | 68.72 | 74.16 | 64.44 | 64.09 | 60.44 | 36.95 | 39.00 | 58.26 |
| | | FP16 | 17.79 | 32.16 | 68.17 | 73.94 | 64.39 | 63.38 | 61.03 | 37.29 | 38.60 | 58.11 |
| | | Diff(↓) | 0.07 | **0.06** | - | - | - | - | - | - | - | 0.14 |
| | MoreauPruner | BF16 | 17.42 | 32.22 | 70.31 | 74.16 | 64.60 | 65.11 | 61.41 | 36.35 | 38.00 | 58.56 |
| | | FP16 | 17.45 | 32.29 | 70.29 | 74.01 | 64.81 | 65.24 | 61.19 | 36.58 | 38.00 | 58.59 |
| | | Diff(↓) | **0.03** | **0.06** | - | - | - | - | - | - | - | **0.03** |
| 15% | LLM-Pruner (Ma et al., 2023) | BF16 | 32.60 | 61.87 | 65.78 | 70.67 | 54.93 | 59.75 | 54.25 | 32.51 | 36.40 | 53.47 |
| | | FP16 | 32.10 | 61.15 | 66.21 | 70.46 | 54.97 | 59.43 | 54.92 | 32.59 | 36.40 | 53.57 |
| | | Diff(↓) | 0.51 | 0.72 | - | - | - | - | - | - | - | 0.10 |
| | MoreauPruner | BF16 | 31.05 | 60.43 | 66.48 | 70.89 | 55.33 | 60.54 | 54.76 | 33.02 | 36.20 | 53.89 |
| | | FP16 | 30.99 | 60.43 | 66.06 | 70.78 | 55.51 | 60.93 | 55.26 | 32.42 | 36.00 | 53.85 |
| | | Diff(↓) | **0.06** | **0.00** | - | - | - | - | - | - | - | **0.04** |
| 20% | LLM-Pruner (Ma et al., 2023) | BF16 | 95.82 | 202.86 | 63.84 | 63.11 | 41.43 | 54.22 | 41.41 | 28.17 | 32.00 | 46.31 |
| | | FP16 | 96.57 | 210.12 | 62.74 | 63.11 | 41.01 | 55.17 | 41.04 | 27.89 | 31.80 | 46.11 |
| | | Diff(↓) | 0.75 | 7.26 | - | - | - | - | - | - | - | 0.20 |
| | MoreauPruner | BF16 | 91.79 | 176.24 | 64.71 | 64.40 | 41.78 | 56.02 | 40.86 | 28.92 | 34.20 | 47.27 |
| | | FP16 | 91.79 | 174.19 | 64.63 | 64.47 | 41.69 | 55.93 | 41.08 | 29.36 | 33.94 | 47.30 |
| | | Diff(↓) | **0.00** | 2.05 | - | - | - | - | - | - | - | **0.03** |

Table 9: Zero-shot performance of the compressed Vicuna-7B.

| Pruning Ratio | Method | WikiText2(↓) | PTB(↓) | BoolQ | PIQA | HellaSwag | WinoGrande | ARC-e | ARC-c | OBQA | QA-Average |
|---|---|---|---|---|---|---|---|---|---|---|---|
| Ratio = 0% | Vicuna-7B† | 16.11 | 61.39 | 76.54 | 77.20 | 70.70 | 67.25 | 65.15 | 41.30 | 40.80 | 62.71 |
| Ratio = 20% w/o finetune | Magnitude† | 3539.98 | 5882.21 | 55.90 | 56.15 | 32.37 | 51.85 | 30.01 | 28.41 | 28.20 | 40.41 |
| | Random† | 34.63 | 112.44 | 61.47 | 70.89 | 54.67 | 56.27 | 55.60 | 31.74 | 34.60 | 52.18 |
| | LLM-Pruner*(Ma et al., 2023) | 25.74 | **92.87** | **61.62** | 74.76 | 63.76 | 56.20 | 63.22 | 36.69 | 37.00 | 56.18 |
| | SmoothGrad | 25.99 | **92.87** | 60.73 | 74.97 | 63.75 | 54.22 | 64.90 | 37.03 | 37.60 | 56.17 |
| | MoreauPruner | **25.54** | 94.34 | 56.82 | **75.79** | 64.35 | 56.35 | **65.95** | **37.88** | **39.80** | **56.76** |
| | MoreauPruner-GS | 30.69 | 108.16 | 61.47 | 75.24 | **66.56** | **61.72** | 57.24 | 37.12 | 38.00 | **56.76** |
| Ratio = 20% w/ finetune | LLM-Pruner*(Ma et al., 2023) | 19.47 | 72.33 | 64.43 | 76.44 | 65.39 | 60.46 | 63.22 | 35.92 | 38.20 | 57.72 |
| | SmoothGrad | 19.51 | **72.05** | 63.46 | 75.68 | 65.38 | 60.93 | 62.79 | 36.43 | 38.80 | 57.64 |
| | MoreauPruner | 19.66 | 73.47 | 63.15 | 76.77 | 65.96 | 60.85 | 65.74 | 37.12 | **40.60** | 58.60 |
| | MoreauPruner-GS | **19.13** | 73.76 | **65.41** | **76.99** | **68.17** | **65.27** | **66.37** | **38.23** | 39.80 | **60.03** |

Table 10: Zero-shot performance of the compressed LLaMA3-8B.

| Pruning Ratio | Method | WikiText2(↓) | PTB(↓) | BoolQ | PIQA | HellaSwag | WinoGrande | ARC-e | ARC-c | OBQA | QA-Average |
|---|---|---|---|---|---|---|---|---|---|---|---|
| Ratio = 0% | LLaMA-8B† | 14.14 | 27.98 | 81.35 | 80.79 | 79.17 | 72.53 | 80.09 | 53.41 | 45.00 | 70.33 |
| Ratio = 20% w/o finetune | LLM-Pruner(Ma et al., 2023) | 25.74 | 45.69 | 67.55 | 74.97 | 63.33 | 67.80 | 62.29 | 35.49 | 36.60 | 58.29 |
| | MoreauPruner-GS | **25.40** | **43.78** | **73.73** | **75.08** | **64.93** | **68.03** | **66.11** | **39.25** | **37.60** | **60.68** |
| Ratio = 20% w/ finetune | LLM-Pruner(Ma et al., 2023) | 23.71 | 42.01 | **77.52** | 77.69 | 71.75 | 67.96 | 71.63 | 42.24 | 40.00 | 64.11 |
| | MoreauPruner-GS | **22.98** | **39.25** | 76.57 | **78.67** | **73.17** | **69.14** | **74.49** | **43.77** | **41.80** | **65.37** |

Table 11: The effect of larger recovery set.

| Pruning Ratio | Method | Recovery Set | BoolQ | PIQA | HellaSwag | WinoGrande | ARC-e | ARC-c | OBQA | AQ-Average |
|---|---|---|---|---|---|---|---|---|---|---|
| Ratio = 0% | LLaMA-7B† | N/A | 73.18 | 78.35 | 72.99 | 67.01 | 67.45 | 41.38 | 42.40 | 63.25 |
| Ratio = 20% | MoreauPruner-GS | 50k(Taori et al., 2023) | 68.87 | 77.26 | 69.81 | 65.04 | 63.64 | 38.23 | 40.60 | 60.49 |
| | MoreauPruner-GS | 2.59M(Wu et al., 2023) | 76.97 | 76.82 | 68.51 | 66.30 | 70.88 | 41.89 | 40.80 | 63.17 (**+2.68**) |

Table 12: Comparison between scratch-training and LLaMA-3B obtained by MoreauPruner-GS

| Method | #Param | BoolQ | PIQA | HellaSwag | WinoGrande | ARC-e | ARC-c | OBQA | QA-Average |
|---|---|---|---|---|---|---|---|---|---|
| StableLM-3B† | 3.6B | 48.78 | **69.48** | 44.52 | 54.62 | 50.93 | 25.17 | 27.40 | 45.84 |
| MoreauPruner-GS | 3.5B | **62.26** | 68.39 | **49.58** | **55.72** | **50.97** | **30.20** | **35.40** | **50.36** |

# E EXPERIMENT DETAILS

## E.1 A DETAILED COMPARISON OF METHODS

We list the comparison on the experiment setting utilized in our baselines, which can be found in Table 13. We should note that the strong competitor LoRAPruning(Zhang et al., 2023) employs an iteratively pruning style, which allows algorithms gradually remove less important weight during

Table 13: A detailed comparison between methods.

| Method | Pruning Criterion | Calibration Set (Size) | Post-Training Set (Size) | Iteratively Pruning | Smoothness |
|---|---|---|---|---|---|
| Random | random | N/A | N/A | ✗ | ✗ |
| Magnitude | $\|\boldsymbol{w}_i\|_2$ | N/A | N/A | ✗ | ✗ |
| WANDA(Sun et al., 2023) | $\|\boldsymbol{w}^{(k)}\|_1\|x_i\|_2$ | C4(0.128k) | C4(20k) | ✗ | ✗ |
| LoRAPrune(Zhang et al., 2023) | $\left\|(\text{LoRA-guided } \frac{\partial L(\boldsymbol{w},\mathcal{D})}{\partial \boldsymbol{w}^{(k)}})\boldsymbol{w}^{(k)}\right\|_1$ | C4(20k) | C4(20k) | ✓ | ✗ |
| LLM-Pruner(Ma et al., 2023) | $\left\|\frac{\partial L(\boldsymbol{w},\mathcal{D})}{\partial \boldsymbol{w}^{(k)}}\boldsymbol{w}^{(k)}\right\|_1$ | Bookcorpus(0.01k) | Alpaca(50k) | ✗ | ✗ |
| SmoothGrad | $\mathbb{E}_{\boldsymbol{z}}\left\|\frac{\partial L(\boldsymbol{w}+\boldsymbol{z},\mathcal{D})}{\partial \boldsymbol{w}^{(k)}}\boldsymbol{w}^{(k)}\right\|_1$ | Bookcorpus(0.01k) | Alpaca(50k) | ✗ | ✓ |
| MoreauPruner | $\|\text{MG}^\rho[g](\boldsymbol{w}) \odot \boldsymbol{w}\|_1^{(k)}$ | Bookcorpus(0.01k) | Alpaca(50k) | ✗ | ✓ |
| MoreauPruner-GS | $\|h\text{-MG}^\rho[g_\sigma](\boldsymbol{w}) \odot \boldsymbol{w}\|_1^{(k)}$ | Bookcorpus(0.01k) | Alpaca(50k) | ✗ | ✓ |

multiple rounds of model pruning and is more time-consuming. Our methods and our primary baseline LLM-Pruner utilize one-shot pruning for efficiency.

### E.2 PARAMETERS CHOOSING

In the pruning stage, we randomly pick a batch from BookCorpus (Zhu et al., 2015) with ten 128-token truncated sentences. The batch choice remains the same among LLM-Pruner, SmoothGrad, MoreauPruner, and MoreauPruner-GS in our experiments. Since deep layers and shallow layers are sensitive to pruning, following previous works (Ma et al., 2023), we only prune the middle layers in this stage. For example, when we aim to prune 20% parameters from LLaMA-7B, we remove 25% parameters from layer 4 to layer 30.

For SmoothGrad, we pass the batch to the model 100 times. We utilized a element-wised Gaussian smoothing, *i.e.,* for weight parameter $\boldsymbol{w}^{(k)}$, the intensity of Gaussian is $\sigma = 0.05\|\boldsymbol{w}^{(k)}\|_1$. The smooth gradient is empirically calculated by averaging the inportance scores of each forward pass.

For both MoreauPruner and MoreauPruner-GS, we also apply the element-wise Gaussian smoothing to the model weights during the optimization of the gradient of the Moreau Envelope, as SmoothGrad does. The hyper-parameter $\rho$ is set to 0.05 for MoreauPruner and 0.2 for MoreauPruner-GS. The stepsize $\gamma$ used in the optimization of the gradient of the Moreau Envelope is 1e-3 for MoreauPruner and 2e-4 for MoreauPruner-GS. The hyper-parameter $\eta$ is set to 5e-6 as explained in the main text. We conducted a parameter search on LLaMA-7B to find suitable hyper-parameters.

In the fine-tuning stage, we use the protocol from previous work (Ma et al., 2023) and employ a LoRA with rank ($r=8$). The batch size is 64. The learning rate is 1e-4 for Alpaca (Taori et al., 2023) and 5e-5 for Lamini (Wu et al., 2023). The training length is two epochs for Alpaca and three epochs for Lamini.

## F    FREQUENTLY ASKED QUESTIONS

In this section, we provide answers to frequently asked questions about our work.

▷ **When pruning an existing model, the parameters are fixed. Why is it meaningful to consider the robustness of pruning criteria against weight perturbation?**

Intuitively, the basic goal of pruning is to remove unnecessary weights while retaining the essential ones. It is reasonable to expect that a pruning algorithm should produce similar results for a model in BF16 and FP16 formats, as models in these formats generally yield the same inference results. The unexpected discrepancy between the two pruned models indicates a bias in the evaluation of weight importance. Therefore, ensuring pruning consistency under minor perturbations can serve as a guiding principle during the design phase of pruning criteria. The improvements in the performance of pruned models, as shown in our numerical results, further support this argument.

▷ **There are many structured pruning methods in the literature. Why does MoreauPruner only compare several of them in this paper?**

The main motivation behind MoreauPruner is to draw attention to the robustness of pruning criteria against weight perturbation. Given that pruning technologies have been extensively explored over the past thirty years, there is significant variation in experiment settings across different methods. These include variations in selected models, calibration/recovery sets, recovery methods, iterative versus one-shot pruning, evaluation metrics/datasets, and more, making it difficult to identify a universally optimal setting.

Therefore, in our paper, we adopt the setting from a recent, powerful, gradient-based baseline, LLM-Pruner, and focus on comparing our method with several competitors that utilize similar experimental setups to ensure a clear comparison. The numerical results and conclusions presented in the main text support our motivations under the selected settings. Our theoretical analysis holds across settings that satisfy our assumption.

▷ **Quantization technologies, another widely-used model compression method, can achieve high compression ratios ($\geq 50\%$) without significant performance drops. Why do we still need pruning methods like MoreauPruner, which typically maintain performance only at lower pruning ratios?**

Pruning and Quantization, as two mainstream model compression methods, are developed for different purposes and each has its own advantages and disadvantages. In practical applications, the choice of compression method should consider several factors. For example, mainstream hardware does not support arbitrary data formats. In cases where an 8-bit model needs to be compressed by approximately 12.5%, pruning would often be a better choice than attempting to quantize the model to 7 bits. Each method serves specific needs, and pruning remains a viable option when fine control over compression is required.

