# OpenReview forum: "MoreauPruner: Robust Structured Pruning of Large Language Models against Weight Perturbations"
_ICLR.cc/2025/Conference — ICLR 2025 Conference Withdrawn Submission_

### Official Review · Reviewer_UeGJ · 2024-10-31

**Soundness:** 2
**Presentation:** 3
**Contribution:** 1
**Rating:** 3
**Confidence:** 4

**Summary:**

The paper introduces MoreauPruner, a structured pruning algorithm designed to enhance the robustness of gradient-based pruing against weight perturbations, LLMs. Traditional gradient-based pruning algorithms often produce unstable results when model weights are slightly altered. To address this issue, MoreauPruner uses the Moreau envelope to stabilize the gradient calculations, ensuring that minor perturbations do not significantly impact pruning outcomes. Through experiments, MoreauPruner demonstrates better robustness and accuracy preservation compared to the baseline methods in ppl and zero-shot accuracies.

**Strengths:**

1. The paper is well-written and concise, and accessible.
2. This work addresses the robustness challenges in gradient-based pruning metrics, presenting a solution validated through experiments.

**Weaknesses:**

1. The paper’s novelty appears limited, as Section 4 on MoreauPruner substantially overlaps with prior work [1]. For instance, Definitions 1, 2, and 3 closely resemble Definitions 1, 2, and 4 in [1]. Additionally, Lines 1-9 in Algorithm 1 are identical to Algorithm 1 in [1].
2. While this method enhances the robustness of gradient-based pruning metrics against weight perturbation (as seen in Table 1), it yields higher perplexity (PPL) at low pruning ratios, such as 5%, and shows only marginal improvements at higher pruning ratios.3.
3. The paper omits several important state-of-the-art baselines in structured pruning for LLMs, including methods presented in [2], [3], and [4].
4. The proposed method offers improvements specifically for gradient-based pruning approaches, which limits its broader applicability. However, gradient-based pruning may not be a competitive option due to its higher computational costs for gradient calculations and relatively limited performance compared to methods in [2], [3], and [4] for LLMs.

[1] Jingwei Zhang and Farzan Farnia. Moreaugrad: Sparse and robust interpretation of neural networks via moreau envelope. ICCV, 2023.
[2] Yongqi An, Xu Zhao, Tao Yu, Ming Tang, and Jinqiao Wang. Fluctuation-based adaptive structured pruning for large language models. In Proceedings of the AAAI Conference on Artificial Intelligence, volume 38, pp. 10865–10873, 2024
[3] Saleh Ashkboos, Maximilian L Croci, Marcelo Gennari do Nascimento, Torsten Hoefler, and James Hensman. Slicegpt: Compress large language models by deleting rows and columns. arXiv preprint arXiv:2401.15024, 2024.
[4] Xuan Shen, Pu Zhao, Yifan Gong, Zhenglun Kong, Zheng Zhan, Yushu Wu, Ming Lin, Chao
Wu, Xue Lin, and Yanzhi Wang. Search for efficient large language models. arXiv preprint arXiv:2409.17372, 2024.

**Questions:**

Apart from the weaknesses, the following questions need to be addressed:
1. Concerning the reported pruning time of 30 minutes on CPU, for models of what size does this apply? Why do you choose to prune on CPUs, not GPUs? Additionally, does this time include the hyperparameter search process as stated?

**Details Of Ethics Concerns:**

The key methodology in this manuscript is nearly identical to that in [1]. For example, Section 4 on MoreauPruner substantially overlaps with prior work [1]. For instance, Definitions 1, 2, and 3 closely resemble Definitions 1, 2, and 4 in [1]. Additionally, Lines 1-9 in Algorithm 1 are identical to Algorithm 1 in [1].

[1] Jingwei Zhang and Farzan Farnia. Moreaugrad: Sparse and robust interpretation of neural networks via moreau envelope. ICCV, 2023.

---

### Official Review · Reviewer_H6p8 · 2024-11-02

**Soundness:** 2
**Presentation:** 3
**Contribution:** 2
**Rating:** 5
**Confidence:** 4

**Summary:**

The paper proposes a new structured pruning algorithm for large language models. The idea is to design a saliency score that is more robust to weight perturbations than the usual gradient-based saliency scores. This is done by taking the gradient of the Moreau envelope of the loss, which can be viewed as a smoothed version of the gradient. The target LLM is pruned using this score, and then fine-tuned with LoRA to recover the performance. The experimental results suggest that the proposed MoreauPruner provides a more robust performance w.r.t. the choice of numeric data types, and outperforms LLM-pruner on various datasets.

**Strengths:**

- The key research question of this paper, i.e., finding a pruning mask that is robust to the weight perturbation in the weight, is quite interesting and thought-provoking.

- The idea of Moreau envelope is mathematically well-grounded and is linked directly to the problem considered; I have learned a lot from this submission.

- The paper is generally quite clear, and well-written.

**Weaknesses:**

- The central weakness of this paper is the unclear motivation. In particular, I am even not sure why the inconsistency of the pruning outcomes up to small perturbation is problematic. There can be many different but equally good pruning masks for a neural network. Indeed, the classic structured pruning work by Srinivas and Babu (2015) proposes to remove either neuron A or neuron B (but not both) whenever two neurons are highly similar to each other, as they are considered redundant. Here, we should be indifferent to the choice whether we remove neuron A or neuron B; both gives us a reasonable good choice. If the authors believe that there should be a unique choice of pruning masks, I recommend giving more supporting evidences, e.g., empirical backups.

- Also, it is not clear to me why the results in Table 1 is meaningful. If I understood correctly, it seems like the authors are trying to prove that MoreauPruner a more robust performance on both BF16 and FP16. Here, I am not sure why the "Diff" matters. Perhaps the fact that MoreauPruner achieves smaller perplexity on both cases is what is meaningful, and having a small "diff" may not really have much implications, as working bad on both numeric types can result in small diff as well. Also, I am curious why MoreauPruner seems to consistently work poorer than LLM-Pruner for 5% pruning ratio.

- The performance gain over LLM-Pruner is not very dramatic, which I believe is perhaps linked to the unclear motivation.

- The set of target models is somewhat restricted, as they are constrained to LLaMA family, except for Vicuna-7B. I recommend adding (at least two) experiments on other options, such as Mistral, Qwen, OPT, or BLOOM.

- I am not sure if I missed, but it would be great if authors could make any remarks on the computational costs of obtaining the pruning masks, such as required memory and computations.

- The quality of writing can be improved. For instance, the paper uses the terminology "structural pruning," which I believe should be replaced with much more popular term "structured pruning" (I am aware that the term "structural pruning" has been used in some recent works, but I think they are mostly mistakes). Also, "anti-intuitive" can be replaced by "counterintuitive." The phrase "robustness pruning result" in L77 is weird. In L165, "easy obtained" should be "easily obtained." In L181, "effiently" should be "efficiently." In L230, "envelop" should be "envelope," as in all other usages. In figure 4 caption, "hyper-paramter" is wrong. There should be much more errors; please locate them and fix them all.

**Questions:**

- A mild suggestion is to draw motivation for perturbation robustness from other sources of weight perturbation. For example, here are some half-baked thoughts: (1) Pruning itself perturbs the weights; that is, pruning some weights may affect gradient evaluations for other weights. (2) Further re-training with LoRA may also be the source of expected weight perturbation; the proposed method can be viewed as being ready for such updates.

---

### Official Review · Reviewer_iFXc · 2024-11-03

**Soundness:** 2
**Presentation:** 3
**Contribution:** 2
**Rating:** 5
**Confidence:** 4

**Summary:**

This paper introduces MoreauPruner, a structured pruning algorithm for LLMs designed to ensure robustness against weight perturbations. The method uses the Moreau envelope to estimate weight importance, aiming to make pruning outcomes more stable even with minor format-based weight discrepancies (e.g., between bfloat16 and float16). The authors provide theoretical support and empirical results, comparing MoreauPruner to existing pruning techniques on LLMs around the 7 billion parameter range, such as LLaMA-7B and Vicuna-7B.

**Strengths:**

1. The paper is generally well-written and easy to follow.

2. The proposed method is empirically shown to be effective for handling parameter perturbation in LLM pruning.

**Weaknesses:**

1. The theoretical contributions and the use of Moreau envelopes in this paper show significant overlap with prior work on MoreauGrad​ [1]. Given these similarities, the novelty of the theoretical contributions is in question, and the approach could be seen as an incremental adaptation rather than a groundbreaking innovation.

2. The experiments focus on relatively small LLMs (~7B parameters). In today’s landscape, where models often exceed 70B parameters, it would be more impactful to evaluate MoreauPruner on larger-scale LLMs to demonstrate its utility and effectiveness in real-world applications.

3. While the authors justify the importance of robustness to weight perturbations in cases like bfloat16 and float16 transitions, this seems limited to niche scenarios. The need for such robustness in broader practical settings remains unconvincing, despite the improvements shown in performance. Expanding on why this aspect is critical for widespread application would strengthen the paper.

[1] Zhang, Jingwei, and Farzan Farnia. "Moreaugrad: Sparse and robust interpretation of neural networks via moreau envelope." Proceedings of the IEEE/CVF International Conference on Computer Vision. 2023.

**Questions:**

1. Can the authors clarify how MoreauPruner would perform with models significantly larger than 7B parameters?

2. Could the approach benefit from more generalized perturbation testing beyond format-specific perturbations?

3. How does the computation cost of MoreauPruner compare to other robust pruning methods, particularly on models where computational efficiency is crucial?

---

### Official Review · Reviewer_LB64 · 2024-11-04

**Soundness:** 3
**Presentation:** 2
**Contribution:** 3
**Rating:** 5
**Confidence:** 4

**Summary:**

This paper shows that one-shot gradient pruning algorithms could lead to unstable results under perturbations to model weights. The authors propose an optimization method MoreauPruner with provable robustness against weight perturbations. The authors conduct experimental evaluation on LLaMA-1 and Vicuna model families.

**Strengths:**

- The authors conduct thorough experiments and ablation studies to demonstrate the effectiveness of the proposed approach.
- This paper looks into an interesting problem as to the instabilies of gradient based pruning method, e.g., across data precision formats. This raises the awareness of the pruning community on the importance of pruning stability.

**Weaknesses:**

- Theorem 1 and 2 are unrelated to the paper. The method is an empirical pruning method, and deriving theoretical bound for Moreau envelope is irrelevant.
- The introduction of section 4.1 seems abrupt. There is little motivation as to why utilizing Moreau-Yosida regularization is helpful.
- The performance improvement on some models seems marginal compared to previous methods, e.g., Table 2 on LLaMA-13B.
- The authors claimed that minor changes in model weights can lead to unstable pruning results of LLMs. However, I don’t feel that Table 1 supports the claim of the paper. From Table 1, it seems that in most cases, the perplexity difference between different number format is minimal, except pruning ratio 20% on PTB datasets.

**Questions:**

Could the authors provide results on more recent LLM model families, e.g. LLaMA-3.1 or Gemma-2?

---

### Official Review · Reviewer_aeR9 · 2024-11-05

**Soundness:** 3
**Presentation:** 4
**Contribution:** 3
**Rating:** 6
**Confidence:** 3

**Summary:**

This paper introduces the MoreauPruner, a robust LLM pruning algorithm that is less sensitive to weight perturbations. The authors provide theoretical insights into the robustness of the pruned model due to properties of the Moreau envelope and demonstrate that their method achieves similar performance to previous comparable methods while being more robust to perturbations. Additionally, the authors present a group sparse pruner and a Smooth Grad variant, along with extensive results on the perplexity and QA performance of their pruned models.

**Strengths:**

The paper is well written and presented. The idea of using the Moreau Envelope is clever and provides some theoretical insight into its impact on pruning. Proper experimental results are provided to back up the claims.

**Weaknesses:**

I have some concerns regarding the Soft Grad and GS pruners, as I am not entirely sure how they enforce sparsity. I would appreciate further explanations on this aspect.

As for the experimental results, it seems fair to say that the pruner does not achieve significant improvements over previous competing methods (as seen in Tables 2–6). Perhaps the more impactful result is its resilience to weight perturbations. I also wonder if there are better downstream tasks than QA to capture the full impact of this pruning method.

I will elaborate on my concerns further in the questions section.

**Questions:**

1. Regarding the setup for changing the precision level, could the authors provide additional references? Or is this a novel evaluation method for the pruning model? Could the authors also conduct more fine-grained experiments concerning floating-point precisions?

2. Is the GST function effectively a soft L1 regularizer?

3. Are there any concerns about the calibration dataset issue mentioned in line 158? If I understand correctly, this is a general assumption in most pruning techniques, correct?

4. Have the authors considered even higher rates of parameter pruning, closer to 70–90%, as attempted by some recent pruning algorithms?

5. Could I get clarification on the evaluation of finetuned versus non-finetuned models?

---

### Author Response · Authors · 2024-11-27

We sincerely thank the Area Chairs and Reviewers for their thorough and constructive feedback on our paper, MoreauPruner. Your insights have been invaluable in helping us identify areas for improvement.

We acknowledge that the paper can be enhanced, particularly in the following aspects:

- Strengthening the experimental results.
- Clarifying the motivation and framing of our contributions.

Given the limited time available to address all the raised concerns comprehensively, we have decided to withdraw the paper to allow for further substantial revisions based on your comments.

We would also like to take this opportunity to address a specific concern regarding the theoretical contributions of MoreauPruner. Our contributions are twofold:

`Robustness to Weight Perturbation:`
To the best of our knowledge, we are the first to explicitly address the robustness of pruning algorithms to weight perturbations. As cited in line 206, "Following [1] …", we introduced the concept of MoreauGrad into the design of pruning criteria. The proposed MoreauPruner integrates the MoreauGrad, model weights, weight structure and group-level sparsity (detailed in Section 4.2), distinguishing our work from [1].

`Generalization of the Moreau Envelope:`
As stated in line 230, "we generalize [1]'s definition…", we extend the concept of the (group) sparse-Moreau envelope in [1] to the $h$-Moreau envelope, where $h$ is any convex function (Definition 3). We provide theoretical results on the robust properties of the $h$-Moreau envelope (Theorem 2) and derive the explicit expression of the $h$-MoreauGrad in the appendix. Our formulation encompasses $\ell_1$-norm and $\ell_{2,1}$-group-norm as special cases of the defined $h$-function, making the results in [1] a subset of our theoretical framework.

We are grateful for the opportunity to improve our work and will incorporate the thoughtful feedback provided as we prepare a revised submission in the future.

**Reference:**

[1] Zhang, Jingwei, and Farzan Farnia. "MoreauGrad: Sparse and robust interpretation of neural networks via Moreau envelope." Proceedings of the IEEE/CVF International Conference on Computer Vision, 2023.

---

### Note · Authors · 2024-11-27

I have read and agree with the venue's withdrawal policy on behalf of myself and my co-authors.